# A Study on the Production of Anhydrous Neodymium Chloride through the Chlorination Reaction of Neodymium Oxide and Ammonium Chloride

**Joo-Won Yu and Jei-Pil Wang ***

Department of Metallurgical Engineering, Pukyong National University, Busan 48513, Republic of Korea; yujw0304@gmail.com
* Correspondence: jpwang@pknu.ac.kr

**Abstract:** The chlorination mechanism of neodymium oxide for the production of anhydrous neodymium chloride was analyzed based on the reaction temperature and reaction ratio of ammonium chloride, considering the suppression of the generation of NdOCl, an intermediate product of the reaction process. The results were obtained by distinguishing the shape of the produced $NdCl_3$ (powder and bulk) and the setup of the chlorination equipment, reflecting its sensitivity to moisture and oxygen. The powdered form of $NdCl_3$ produced at 400 °C and under argon gas flow was identified as $NdCl_3 \cdot 6(H_2O)$, while the bulk form of $NdCl_3$ produced by melting at 760 °C after a chlorination process consisted of anhydrous $NdCl_3$ and $NdCl_3 \cdot n(H_2O)$. The powdered $NdCl_3$ produced in an argon gas environment with a controlled level of oxygen (below 16.05 ppm) and moisture (below 0.01 ppm) content was identified as single-phase anhydrous $NdCl_3$ and showed the highest chlorination conversion rate of 98.65%. The addition of overstoichiometric ratios of $NH_4Cl$ in the chlorination process decreased the total amount of impurities (N, H, and O) in the $NdCl_3$ product and increased the conversion rate of $NdCl_3$.

**Keywords:** chlorination process; neodymium oxide; ammonium chloride; anhydrous neodymium chloride (lll)

## 1. Introduction

Driven by global concerns about climate change, policies are being strengthened to reduce greenhouse gas emissions from industrial processes, and research is actively underway to replace carbon-based processes. The electric vehicle market in the automotive industry is growing rapidly to achieve the national carbon neutrality goals set by the 2015 Paris Agreement [1]. This trend is also affecting the metal market, increasing the demand for neodymium–iron–boron (NdFeB) permanent magnets, an essential component of electric vehicle motors, and the development of efficient and environmentally friendly production processes is in demand [2,3].

Neodymium metal, a key component of permanent magnets, is produced by removing oxygen from $Nd_2O_3$ (Neodymium oxide) powder obtained from the physical and chemical treatment of rare earth minerals or industrial waste, such as bastnaesite and monazite. The electrowinning process, a commercialized rare earth smelting method in China, uses the heat released during electrolysis and molten salts to produce metal from metal oxides and has the advantage of being able to operate continuously in an atmospheric environment. However, the generation of $CO/CO_2$ gas due to the use of graphite anodes and the formation of harmful gases such as $CF_4$ and $C_2F_6$ generated from the electrolyte during operation require improvement in terms of its environmental impact [4,5]. A method used for the chlorination of rare earth oxides using imidazolium chloridoaluminate ILs, such as $[BMIm]Cl \cdot nAlCl_3$, has been reported [6]. This method has the advantage of being able to prepare anhydrous $RECl_3$ (rare earth chloride) at a relatively low temperature of 175 °C

using less toxic materials. Unfortunately, this method requires a relatively long processing time (>24 h) and subsequent washing with toluene and DCM to purify the product after chlorination. In addition, there was an issue with the residual washing solution in the sample. These factors can reduce the production efficiency in continuous processes, where chlorinated products are used as raw materials (Nd sources) in metal production steps. Among the carbon-free smelting processes, metallothermic reduction, which produces metals through the direct reaction of metal oxides with reducing agents such as Ca and Mg, offers the advantage of reduced operational steps owing to the simplified process steps and the possibility of producing products in the form of alloys, which also allows for reduced process temperatures. However, the generation of slag via the combination of a reducing agent and oxygen during the reaction process requires additional processes to recycle the slag after operation [7]. One way to prevent slag generation is to convert metal oxides into metal salts through chlorination or fluorination reactions and use them as raw materials (Nd sources) for metal manufacturing. The mechanism of chlorination and fluorination reactions, based on thermal energy, involves the use of a reducing agent or catalyst to convert oxygen bound to metal oxides into gaseous compounds and to combine metal elements with chlorine or fluorine to form metal salts. Among the studies reported on the chlorination and fluorination of metal oxides [8–11], the carbochlorination process used to produce neodymium chloride ($NdCl_3$) offers the advantages of low reaction temperatures and formation of anhydrous chloride because of the use of carbon as a reducing agent. However, the use of carbon and chlorine gas can lead to greenhouse gas generation and reactor corrosion, and in some cases, the intermediate products of the chlorination reaction, neodymium oxychloride (NdOCl) and $NdCl_3 \cdot n(H_2O)$ with crystallization water, can be produced [8].

Another method for the chlorination of $Nd_2O_3$ is the use of ammonium chloride ($NH_4Cl$) as the chlorinating agent. In the chlorination of $Nd_2O_3$ using $NH_4Cl$, the hydrogen in ammonium chloride combines with the oxygen in $Nd_2O_3$ and is removed as a gas, while chlorine combines with neodymium to form $NdCl_3$. $NH_4Cl$ can convert $Nd_2O_3$ into $NdCl_3$ at a relatively low temperature and does not generate greenhouse gases or slag during this process [9]. However, research on the chlorination of $Nd_2O_3$ using $NH_4Cl$ is lacking.

Therefore, this study aimed to elucidate the thermodynamic mechanism of the chlorination process between $Nd_2O_3$ and $NH_4Cl$ and to derive the preparation conditions for anhydrous $NdCl_3$. In addition, the control of impurity concentration was investigated as a way to suppress the formation of NdOCl and $NdCl_3 \cdot n(H_2O)$ phase in the chlorination process for the preparation of anhydrous $NdCl_3$.

## 2. Materials and Methods

### 2.1. Materials

The calcination process of $Nd_2O_3$ (Alfa Aesar, 99%) for removing impurities prior to the chlorination process was investigated using simultaneous DSC-TG (differential scanning calorimetry-thermogravimetric analysis) measurements and XRD (X-ray diffraction) analysis.

Figure 1 shows the DSC-TG curves of raw $Nd_2O_3$ powder. Three endothermic peaks and stepwise mass losses were observed during the heating process at temperatures up to 1200 °C in a nitrogen atmosphere. The first endothermic peak observed at 95.38 °C is expected to be due to the evaporation of moisture present in the $Nd_2O_3$ powder, while the subsequent endothermic peaks measured around 270.49 °C and 411.67 °C were confirmed to be caused by the phase transformation of $Nd(OH)_3$. The stepwise phase transformation of $Nd(OH)_3$ is as follows [12,13]:

$$Nd(OH)_{3(s)} \rightarrow NdOOH_{(s)} + H_2O_{(L)} \ (T > 270\ ^\circ C) \tag{1}$$

$$2NdOOH_{(s)} \rightarrow Nd_2O_{3(s)} + H_2O_{(L)} \ (T > 430\ ^\circ C) \tag{2}$$

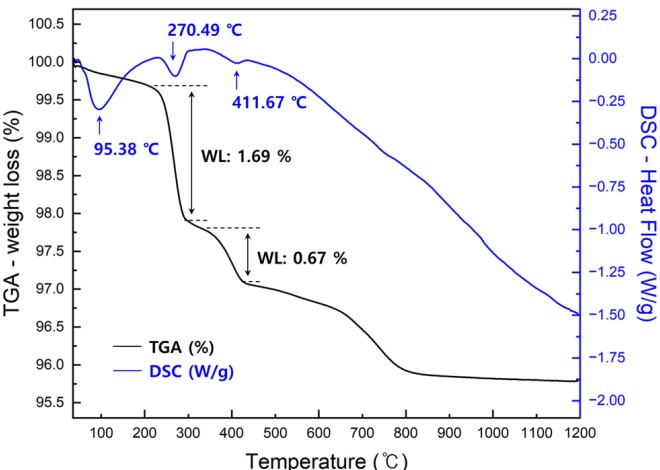

**Figure 1.** DSC-TG curves of raw neodymium oxide measured under a nitrogen atmosphere with a heating rate of 5 °C/min from room temperature to 1200 °C.

The mass loss of the sample due to the evaporation of $H_2O_{(g)}$ during the phase transition was found to be 1.69 wt.% at 270.49 °C and 0.67 wt.% at 411.67 °C. The phase transition temperature of NdOOH showed a deviation of about 19 °C compared to Refs. [12,13]. The total mass reduction of the sample through the stepwise reaction was found to be 4.21 wt.%.

Figure 2a presents the X-ray diffraction (XRD) pattern of $Nd_2O_3$ before calcination, whereas Figure 2b illustrates the phase composition of the $Nd_2O_3$ powder dried at 850 °C for 2 h in an Ar atmosphere. The calcination process confirmed the transformation of $Nd(OH)_3$ present in $Nd_2O_3$ into the cubic $Nd_2O_3$ phase. Therefore, the $Nd_2O_3$ powder used in this study was subjected to a drying process in an argon atmosphere at 850 °C for 2 h prior to the chlorination experiment.

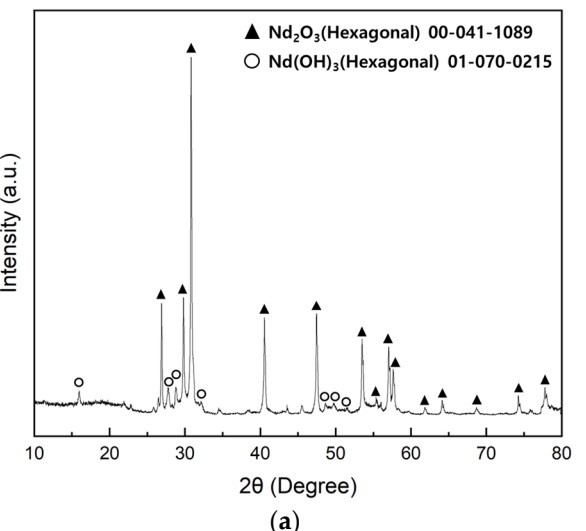

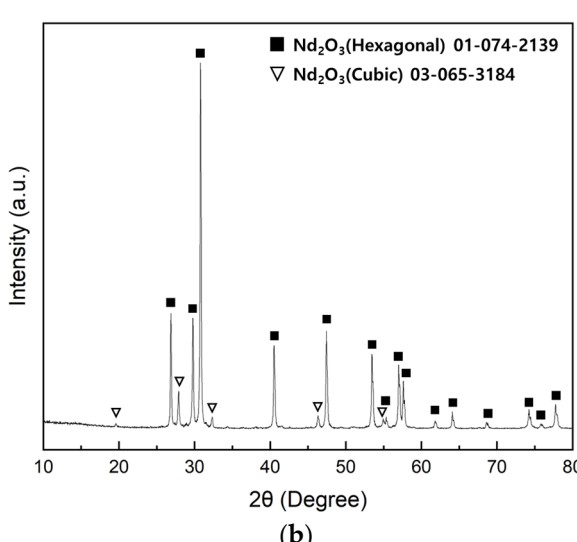

**Figure 2.** XRD patterns of $Nd_2O_3$ before and after calcination: (**a**) Raw material (**b**) After calcination at 850 °C for 2 h in Ar atmosphere.

Thermodynamic analysis was performed to derive a drying temperature that considered the sublimation point to remove residual moisture in the $NH_4Cl$ powder (JUNSEI, 98.5%) used in the chlorination experiment. To satisfy the diffusion equilibrium condition $d\mu_s(T, P) = dM_v(T, P)$ between the solid and gas phases of a single-component system, the Clausius–Clapeyron (C-C) equation was applied under the condition that $NH_4Cl_{(g)}$

does not dissociate into $NH_{3(g)}$ and $HCl_{(g)}$ in the gas phase. The equilibrium vapor pressure of $NH_4Cl$ according to the C-C relationship is expressed as shown in Equation (3) [14]:

$$\ln\left(\frac{P}{Torr}\right) = 33.043 - 10868.3\left(\frac{K}{T}\right) - 1.34\ln\left(\frac{T}{K}\right)$$
$$(Latent\ heat\ of\ sublimation:\ \Delta h^o_{v/s}\ (550\ K) = 10868.3\ J/mol)$$

(3)

Figure 3 represents the equilibrium vapor pressure curve of $NH_4Cl$ for each temperature calculated from the C-C relationship in Equation (3), and it was confirmed that pure $NH_4Cl$ has a vapor pressure of 760 Torr at 338 °C. A visible increase in vapor pressure was observed from approximately 200 °C, and the thermodynamic analysis confirmed that pure $NH_4Cl$ underwent a phase transition from the CsCl structure to the NaCl structure at a temperature of 184.5 °C. Thus, the $NH_4Cl$ powder used in the chlorination experiment was dried at 150 °C for 2 h in an Ar atmosphere prior to the chlorination experiment.

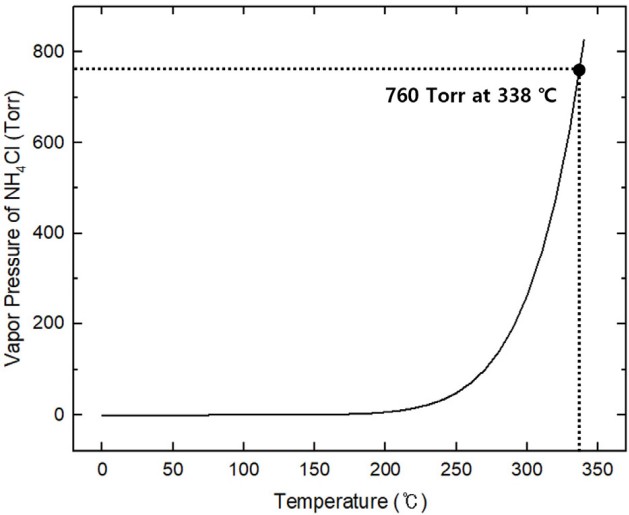

**Figure 3.** Vapor pressure curve of $NH_4Cl$ by temperature.

Table 1 represents the estimated reaction temperature of ceramic-based crucibles and materials (reactants and products) in the chlorination process at a temperature of 1 atm, 1000 °C or below. Thermodynamic analyses were conducted using the FactPS, FTfrtz, and FToxid databases in FactSage 8.2, and chlorination experiments were performed using ideal quartz crucibles.

**Table 1.** Reaction temperatures with substances (reactants and product) for each crucible at temperatures below 1000 °C.

| Crucible | $Nd_2O_3$ | $NH_4Cl$ | $NdCl_3$ |
|---|---|---|---|
| MgO | Stable | 0~1000 | 0~1000 |
| $Al_2O_3$ | Stable | 500 | 300 |
| $SiO_2$ | Stable | Stable | 700 |

## 2.2. Experimental Apparatus

Figure 4 shows a schematic of the horizontal tube furnace with the argon gas flow atmosphere used in this study. The reaction tube consisted of a quartz tube (L600 mm × D49 mm × T2 mm) and a water-cooled SUS covered with an O ring at both ends. A quartz tube was used to prevent corrosion caused by salt in the chlorination reaction area. The Ar gas (>99.99%) line consisted of a flow meter (max: 500 mL/min) to regulate the gas supply and a check valve to prevent corrosion. An R-type thermocouple coated with alumina was used to prevent salt corrosion. The area connected to the metal

cover was sealed with Teflon tape to prevent leakage and then coupled with an O-ring. The gas outlet line consisted of an Erlenmeyer flask to collect fumes and a neutralization chamber to collect the gas products.

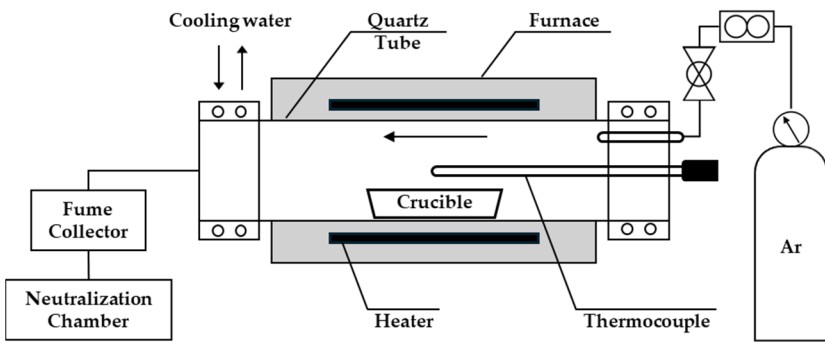

**Figure 4.** Schematic diagram of the horizontal tube furnace used in the chlorination experiment.

Figure 5 shows a schematic of the experimental setup in a glove box filled with Ar gas. The main compartment of the glove box was primarily purged by Ar gas, and $O_{2(g)}$ and $H_2O_{(g)}$ concentrations were controlled at an average of 16.05 ppm and 0.01 ppm, respectively. The heater was attached to the reactor compartment located at the bottom of the main compartment of the glove box. A thermocouple was attached to the reactor crucible. The raw materials were maintained in an antechamber under vacuum ($1 \times 10^{-5}$ bar) for 6 h to remove the air present in the powder and then charged into the reactor. The line was attached to the top of the heater and connected to an Erlenmeyer flask to collect the fumes generated during the reaction.

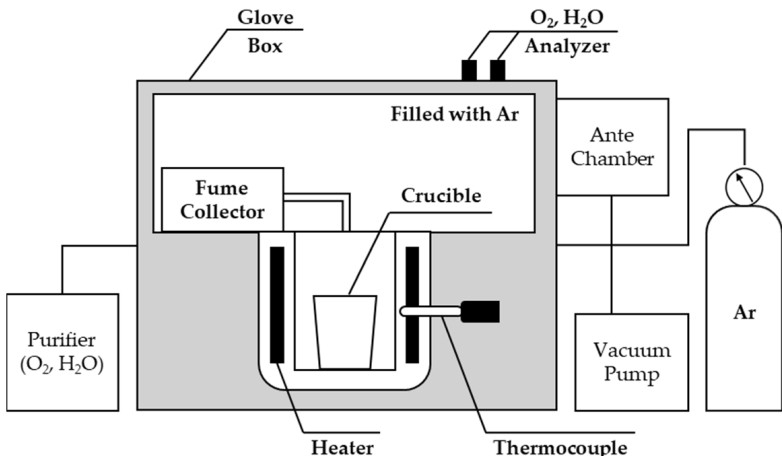

**Figure 5.** Schematic diagram of the glove box used in the chlorination experiment under an Ar atmosphere.

### 2.3. Experimental Analysis

The phases of $Nd_2O_3$ before and after calcination and those of Nd chlorides prepared in a horizontal tube furnace were confirmed via X-ray diffraction (XRD) analysis. The measurements were performed using a Rigaku UltimaIV powder diffractometer (Tokyo, Japan) with CuKα1 radiation (λ = 0.154060 nm) over a 2θ range of 0–80° at a scan rate of 3°/min. The calcination temperature of $Nd_2O_3$ was investigated using a simultaneous DSC-TG thermal analyzer (TA Instruments, SDT 650, New Castle, DE, USA). The sample was heated from room temperature to 1200 °C at a rate of 5 °C/min under a nitrogen atmosphere. The particle size distribution of the Nd chloride prepared in powder form was confirmed using a laser diffraction particle size analyzer (Mastersizer 3000, Malvern Panalytical Ltd., Malvern, Worcs, UK). Scanning electron microscopy (SEM) images were obtained using

a Schottky microscope (JEOL, JSM-IT800SHL, Tokyo, Japan). The samples were sputter-coated with platinum before examination. The concentrations of impurities (nitrogen, hydrogen, and oxygen) in the Nd chloride were confirmed via elemental analysis. N and H were analyzed using a UNI cube instrument (EA KOREA, Hanam, Korea), and the oxygen concentration was determined using a RAPID OXY cube instrument (EA KOREA, Hanam, Republic of Korea). The phase of $NdCl_3$ produced in the glove box under an Ar atmosphere was confirmed via X-ray diffraction measurements using a "Bruker, D8 Advance A25 Plus, Billerica, MA, USA" diffractometer with CuK$\alpha$1 radiation ($\lambda$ = 0.15406 nm) and an airtight holder (Bruker model). A 2$\theta$ range of 0–90° was scanned at a rate of 3°/min.

*2.4. Experimental Procedure*

2.4.1. Horizontal Tube Furnace Apparatus: Manufacturing of $NdCl_3$ Powder

Dried $Nd_2O_3$ powder and $NH_4Cl$ powder were mixed in the required proportions, placed in a quartz crucible, and placed in the center of a reaction tube. The Ar gas flow rate was maintained at 500 mL/min for 30 min to remove gaseous impurities from the reaction tube. Afterward, the temperature was raised to 400 °C at a heating rate of 5 °C/min under Ar flow. This condition was maintained for 120 and 240 min, depending on the experimental batch, and the furnace was then naturally cooled to room temperature. After the completion of the experiment, the mass of the product was measured and placed in a sealed conical tube to protect the product from moisture. Subsequently, the analyses (XRD, PSA, SEM, and elemental analysis) were performed.

2.4.2. Horizontal Tube Furnace Apparatus: Manufacturing of Bulk $NdCl_3$

The production of $NdCl_3$ in a bulk form via the chlorination reaction reduces the impact of the reaction with moisture contained in the air, owing to the significant reduction in the contact surface of the powder product and moisture contained in the air. The temperature was raised to 400 °C under the same conditions as in the $NdCl_3$ powder manufacturing process and maintained for 75 and 120 min, respectively. Subsequently, it was heated to 760 °C at a heating rate of 5 °C/min under Ar flow, maintained for 1 min at 760 °C, and then the furnace was naturally cooled to room temperature. Minimizing the duration that the product is maintained at 760 °C is necessary due to the complicated separation of the bulk product from the quartz crucible when maintained for more than 1 min. This could potentially affect the recovery rate of $NdCl_3$. After the completion of the experiment, the product was recovered, its mass was measured, the product was placed in a conical tube, and the tube was sealed to protect it from moisture. The bulk products were pulverized into a powder of 45 μm or less for XRD and elemental analysis.

2.4.3. Glove Box Apparatus: Manufacturing of $NdCl_3$ Powder

The raw materials and crucible were placed in a glove box using an antechamber, and each powder was placed in an individual quartz crucible and then dried separately in the reactor area of the glove box. Firstly, $Nd_2O_3$ powder was heated to 850 °C with a heating rate of 5 °C/min, maintained for 2 h, and naturally cooled to room temperature. Secondly, the $NH_4Cl$ powder was heated to 150 °C with a heating rate of 5 °C/min, maintained for 2 h, and then naturally cooled to room temperature. Dried $Nd_2O_3$ powder and $NH_4Cl$ powder were mixed in the required proportions and placed in a quartz crucible, which was placed at the center of the reactor area. Afterward, the reactor was heated to 400 °C with a heating rate of 5 °C/min, maintained for 120 min, and then naturally cooled to room temperature. After the completion of the experiment, the product was recovered, its mass was measured, and some samples were stored in an airtight holder (Bruker model) for XRD analysis. To prepare for the PSA, SEM, and elemental analyses, the sample was stored in a conical tube and opened immediately before analysis to minimize the effects of moisture.

### 2.5. Conversion Rate of Chlorination

The chlorination conversion rate was derived by comparing the theoretical mass of $NdCl_3$ that could be produced from the $Nd_2O_3$ used with the mass of anhydrous $NdCl_3$ obtained experimentally. Elemental analysis of nitrogen (N), hydrogen (H), and oxygen (O) was performed to derive the concentration of impurities present in the product as intermediate products, such as NdOCl, hydrate $NdCl_3 \cdot n(H_2O)$ and residual $NH_4Cl$. The chlorination conversion rate, considering the concentration of impurities, is given by Equation (4):

$$Conversion\ rate(\%) = \frac{[Mass\ of\ sample] - [Mass\ of\ impurities]}{Theoretical\ production\ amount\ of\ NdCl_3} \times 100\% \qquad (4)$$

### 2.6. Chlorination Variables

Since pure $NdCl_3$ undergoes a phase transition to $NdCl_3 \cdot (H_2O)_6$ and NdOCl when reacted with moisture and oxygen at room temperature, respectively, the gaseous atmosphere was formed with Ar during the chlorination process. Considering the contact and reaction areas of the product and air after chlorination, the experiments were conducted by separating the geometries of the apparatus and product. The chlorination temperature and reaction rate were derived from a thermodynamic analysis using FactSage 8.2, and the reaction time was applied starting from 4 h and gradually decreasing. Table 2 lists the chlorination conditions used in this study.

**Table 2.** Experimental conditions.

| No. | Apparatus | Gas Atmosphere | Product Form | Chlorination | | Melting | | Reactants [g] | |
|---|---|---|---|---|---|---|---|---|---|
| | | | | Temp. [°C] | Time [min] | Temp. [°C] | Time [min] | $Nd_2O_3$ | $NH_4Cl$ |
| 1 | Horizontal furnace | Ar gas blowing | Powder | 400 | 240 | - | | 5 | 7.4796 |
| 2 | | | | | | | | | 9.0455 |
| 3 | | | | | | | | | 19.2753 |
| 4 | | | | | | | | | 7.4796 |
| 5 | | | | | 120 | | | | 9.0455 |
| 6 | | | | | | | | | 19.2753 |
| 7 | | | Bulk | 400 | 120 | 760 | 1 | 5 | 7.4796 |
| 8 | | | | | | | | | 9.0455 |
| 9 | | | | | | | | | 19.2753 |
| 10 | | | | | 75 | | | | 7.4796 |
| 11 | | | | | | | | | 9.0455 |
| 12 | | | | | | | | | 19.2753 |
| 13 | Glove box | $O_2$: 16.05 ppm $H_2O$: 0.01 ppm | Powder | 400 | 120 | - | | 5 | 7.4796 |

## 3. Results

### 3.1. Thermodynamic Considerations

3.1.1. Calcination

Thermodynamic analysis confirmed that $Nd(OH)_3$ present in $Nd_2O_3$ reacted with $NH_4Cl$ during chlorination to produce $NdCl_3$ and NdOCl, as shown in Figure 6a. The presence of NdOCl in $NdCl_3$ may result in the use of metal-reducing agents above stoichiometric ratios and additional salts to reduce slag activity in the subsequent Nd metal production steps. Figure 6b shows the equilibrium composition of $Nd(OH)_3$ at different

system temperatures. It was observed that $Nd(OH)_3$ exhibited noticeable decomposition behavior above 800 °C. The thermodynamic calcination of $Nd(OH)_3$ is as follows:

$$2Nd(OH)_{3(s)} \rightarrow Nd_2O_{3(s)} + 3H_2O_{(g)}, \quad T > 800\ °C \tag{5}$$

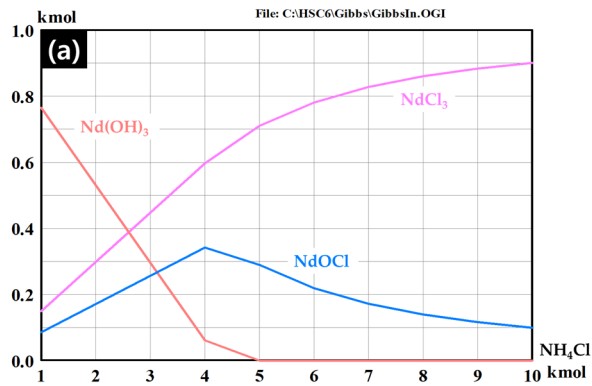 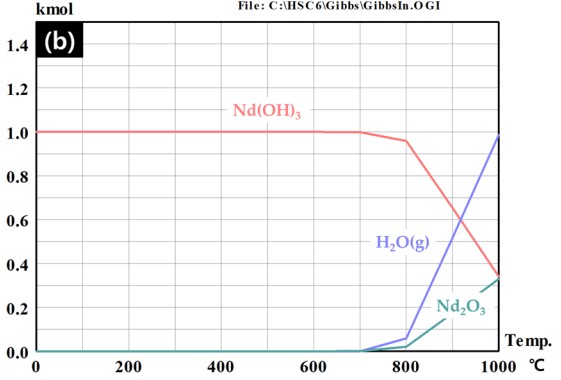

**Figure 6.** (**a**) Equilibrium composition with increasing reaction ratio of $NH_4Cl$ at 400 °C based on 1 mole of $Nd(OH)_3$ (**b**) Equilibrium composition of a system with 1 mole of $Nd(OH)_3$ at different temperatures calculated by HSC chemistry 6.12.

3.1.2. Chlorination

Thermodynamic analysis was conducted using the FactPS and FTfrtz databases in FactSage 8.2 to determine the chlorination temperature and reaction ratio of $Nd_2O_3$: $NH_4Cl$ (mol/mol). Based on Ref. [10], the thermodynamic reaction for the chlorination of $Nd_2O_3$ can be expressed as

$$Nd_2O_{3(s)} + 2NH_4Cl_{(s)} \rightarrow 2NdOCl_{(s)} + 2NH_{3(g)} + H_2O_{(g)}, \quad T < 306.47\ °C \tag{6}$$

$$Nd_2O_{3(s)} + 2NH_4Cl_{(s)} \rightarrow 2NdOCl_{(s)} + 2NH_{3(g)} + H_2O_{(g)}$$
$$NdOCl_{(s)} + 2NH_4Cl_{(s)} \rightarrow NdCl_{3(s)} + 2NH_{3(g)} + H_2O_{(g)}, \quad T = 306.47\ °C \tag{7}$$

$$Nd_2O_{3(s)} + 9.4NH_4Cl_{(s)} \rightarrow 2NdCl_{3(s)} + 9.4NH_{3(g)} + 3H_2O_{(g)} + 3.4HCl_{(g)},$$
$$T = 400\ °C, \Delta G°_{673.15K} = -233.97\ kJ \tag{8}$$

It was confirmed that $NdOCl_{(s)}$, an intermediate product of the chlorination reaction of $Nd_2O_{3(s)}$ and $NH_4Cl_{(s)}$, was converted to $NdCl_{3(s)}$, $NH_{3(g)}$, and $H_2O_{(g)}$ at temperatures above 306.47 °C, as shown in Equation (6). Increasing the system temperature increased the amount of $NH_4Cl_{(s)}$ required to produce $NdCl_{3(s)}$ and promoted the formation of $HCl_{(g)}$. Using Equation (8) as an example, the molar ratio of $NH_4Cl_{(s)}$ required to produce 2 moles of $NdCl_{3(s)}$ increased to 9.4 moles and $HCl_{(g)}$ was formed as the system temperature increased to 400 °C. Excess $NH_4Cl_{(S)}$ contributed to the acidic nature of the reaction process and readily dissociated into $HCl_{(g)}$ and $NH_{3(g)}$, converting $Nd_2O_{3(s)}$ into $NdCl_{3(s)}$ via an acid–base reaction to prevent the hydrolysis of the produced $NdCl_{3(s)}$ [9].

Thermodynamic modelling to derive the reaction ratio of $Nd_2O_{3(s)}$ and $NH_4Cl_{(s)}$ and the chlorination temperature from Equation (8) is shown in Figure 7. Figure 7a shows the reaction entropy with an increasing reaction ratio of $NH_4Cl/Nd_2O_3$ (mol/mol) for each system temperature, and it was confirmed that the entropy change increased rapidly due to the phase transition of oxygen in $NdOCl_{(s)}$ to $H_2O_{(g)}$ at temperatures above 306.47 °C. Figure 7b shows the Gibbs free energy of reaction($\Delta G_{rxn}$) with an increasing reaction ratio of $NH_4Cl/Nd_2O_3$ (mol/mol) for each system temperature. When the system temperature was constant, an increase in the reaction ratio of $NH_4Cl_{(s)}$ leads to a decrease in $\Delta G_{rxn}$, and the corresponding decrease in the Gibbs free energy change was attributed to the increase in gaseous products and the formation of a gaseous solution after the reaction. Figure 7c represents the number of moles of the stable phase as the reaction ratio of

$NH_4Cl_{(s)}$ increased for each system temperature based on 1 mole of $Nd_2O_{3(s)}$, and it was observed that a higher system temperature increased the amount of $NH_4Cl_{(s)}$ required to generate 2 mole of $NdCl_{3(s)}$. Figure 7d illustrates the variation in $NdOCl_{(s)}$ activity in the system with an increasing reaction ratio of $NH_4Cl/Nd_2O_3$ (mol/mol) at different system temperatures. Thermodynamically, the activity of a pure solid is defined as 1. According to the calculation method and assumptions in the FactSage program, when the activity of a pure solid is less than 1, the formation of a solid phase is considered impossible [15]. Therefore, increasing the $NH_4Cl_{(s)}$ molar ratio decreased $NdOCl_{(s)}$ activity, and the system reached equilibrium in a direction that suppressed the formation of the $NdOCl_{(s)}$ phase. In contrast, the $NdOCl_{(s)}$ activity increased at higher temperatures at a fixed $NH_4Cl_{(s)}$ molar ratio. Consequently, increasing the temperature of the system was beneficial for improving the reaction driving force; however, it could also increase the $NdOCl_{(s)}$ activity and the amount of $NH_4Cl_{(s)}$ required for $NdCl_{3(s)}$ formation. Figure 7e shows the activity of the stable phases as the reaction ratio of $NH_4Cl/Nd_2O_3$ (mol/mol) increased, based on 1 mole of $Nd_2O_{3(s)}$ at a temperature of 400 °C, and NdOCl was found to decrease after passing through a coexistence region where it forms a phase mixture with $Nd_2O_{3(s)}$ in the range of about 2.5 to 9.4 mole of $NH_4Cl_{(s)}$. Therefore, to suppress the production of $NdOCl_{(s)}$ in the chlorination reaction, the reaction ratios of $NH_4Cl_{(s)}$ were selected in amounts corresponding to the activity of $NdOCl_{(s)}$ of 0.1, 0.5, and 1. Figure 7f represents the changes in Gibbs free energy according to changes in the activity of $NdOCl_{(s)}$ at a system temperature of 400 °C, and it is confirmed that the change in free energy was decreased at lower activities of $NdOCl_{(s)}$. The points in Figure 7f represent the same states as those in Figure 7e.

The Gibbs free energy (ΔG) in a non-standard state can be obtained with the Van't Hoff reaction isotherm in Equations (9) and (11), which represents the reaction quotient in the chlorination reaction at 400 °C. Due to the formation of a gaseous solution during the reaction, ΔG has a lower value than the standard Gibbs free energy change (ΔG°), which increases the driving force of the chlorination reaction. Table 3 presents the thermodynamic parameters in the non-standard state according to the system temperature and the reaction ratio of $NH_4Cl/Nd_2O_3$(mol/mol).

$$\Delta G = \Delta G^o + RTlnQ \tag{9}$$

where *R*: gas constant (J/mol·K), *T*: absolute temperature (K), *Q*: Reaction quotient

$$K_C = exp\left(\frac{-\Delta G^o}{RT}\right) = 1.43 \times 10^{18} \tag{10}$$

where *Kc*: equilibrium constant at 400 °C

$$Q = \frac{(a_{NdCl_3})^2 \times (P_{NH_3})^{9.4} \times (P_{H_2})^3 \times (P_{HCl})^{3.4}}{(a_{Nd_2O_3})^2 * (a_{NH_4Cl})^{9.4}} \ [Q < 1] \tag{11}$$

where *P*: partial pressure, *a*: activity (pure condensed species in their standard states have an activity of 1).

**Table 3.** Thermodynamic parameters according to the reaction ratio of $Nd_2O_3$: $NH_4Cl$ in chlorination reaction at 400 °C calculated with the FactPS, FTfrtz database in Factsage 8.2 software.

| Temp. [°C] | $NH_4Cl/Nd_2O_3$ [mol/mol] | a(NdOCl) | $\Delta G_{rxn}$ [kJ] | $\Delta H_{rxn}$ [kJ] | $\Delta S_{rxn}$ [J] |
|---|---|---|---|---|---|
| 400 | 9.41 | 1 | −908 | 1524 | 3027 |
| 400 | 11.38 | 0.5 | −1051 | 1927 | 3735 |
| 400 | 24.25 | 0.1 | −1965 | 4549 | 8308 |

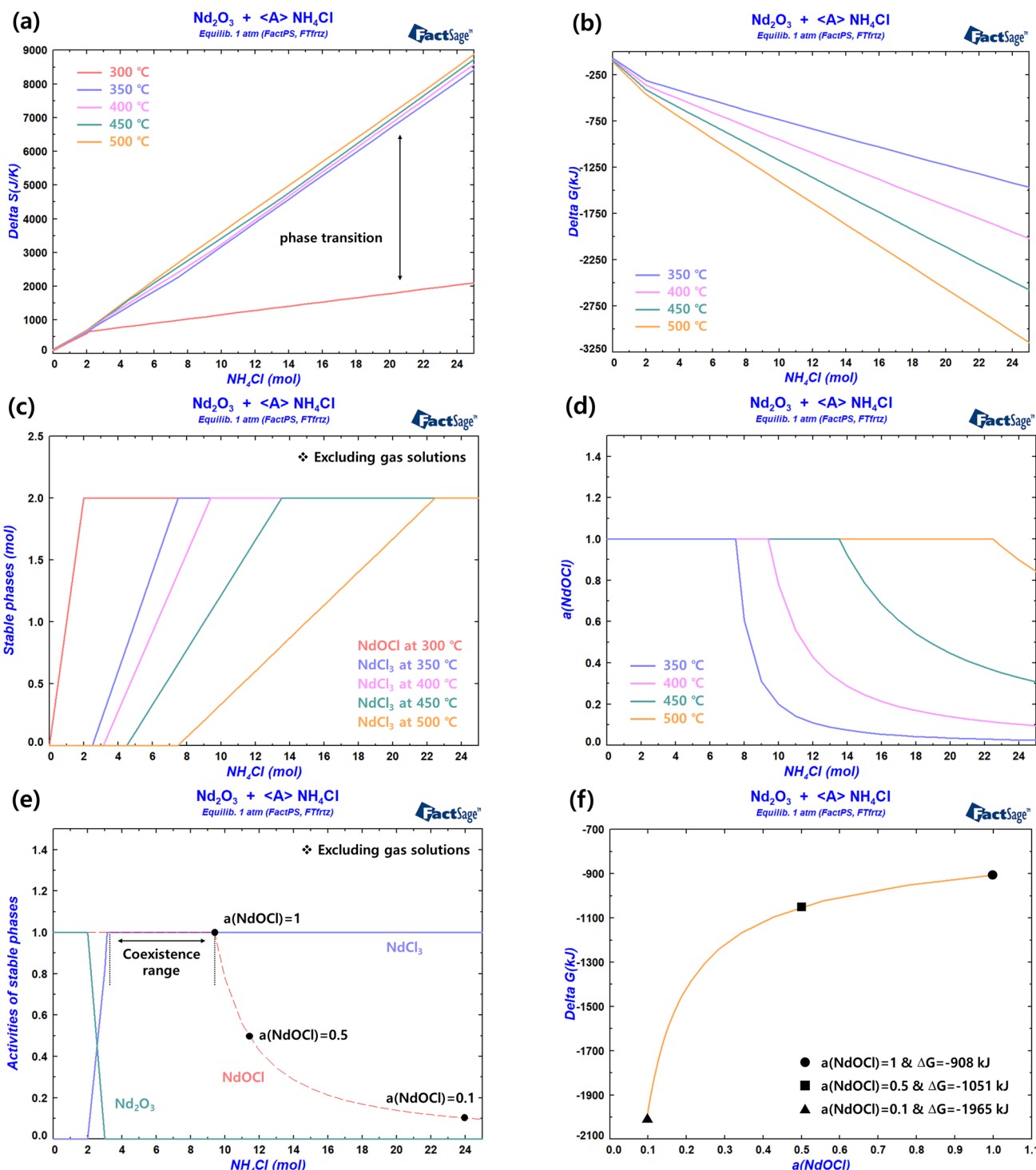

**Figure 7.** Thermodynamic model for the chlorination reaction of $Nd_2O_3$ with $NH_4Cl$ (all plots are calculated based on 1 mole of $Nd_2O_3$): (**a**) The entropy change with increasing reaction ratio of $NH_4Cl/Nd_2O_3$ (mol/mol) for each system temperature. (**b**) The change in Gibbs free energy with increasing reaction ratio of $NH_4Cl/Nd_2O_3$ (mol/mol) for each system temperature. (**c**) The number of moles of stable phases as $NH_4Cl$ reaction ratio increases for each system temperature. (**d**) The change in $NdOCl_{(s)}$ activity in the system with increasing reaction ratio of $NH_4Cl/Nd_2O_3$ (mol/mol) for each system temperature. (**e**) The changes in activity of stable phases in the system as the reaction ratio of $NH_4Cl/Nd_2O_3$ increases at 400 °C. (**f**) The change in Gibbs free energy as the degree of NdOCl activity in the system changes at 400 °C.

### 3.2. Horizontal Tube Furnace Apparatus: Manufacturing of NdCl₃ Powder

Figure 8 shows the phase compositions of the products manufactured at 400 °C with various reaction ratios of $Nd_2O_3$ and $NH_4Cl$ (1:9.41, 1:11.38, 1:24.25 mol/mol) and reaction times (120 and 240 min) as chlorination process parameters. $NdCl_3 \cdot (H_2O)_6$ was detected as the main product in all samples.

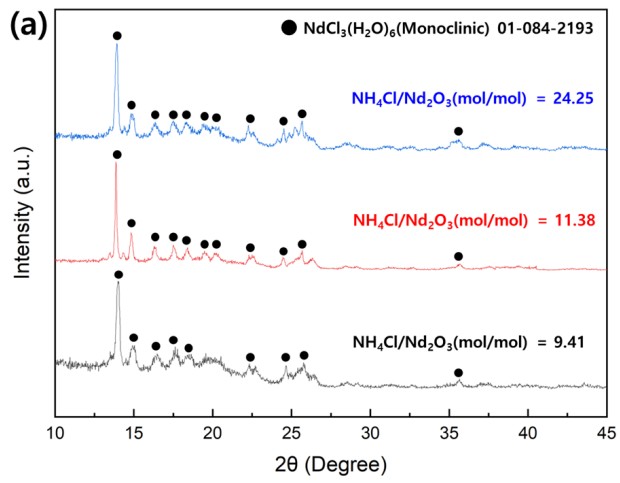 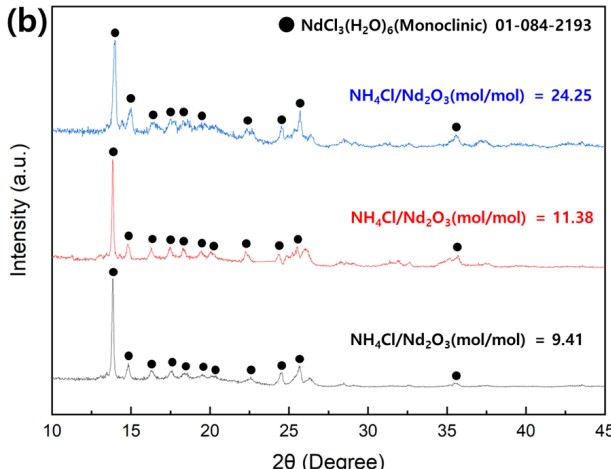

**Figure 8.** XRD pattern of Nd chloride powder ($Nd_2O_3$ to $NH_4Cl$ reaction ratio shown in the figure): (**a**) 120 min chlorination and (**b**) 240 min chlorination.

Particle size analysis was conducted to investigate the effects of the $Nd_2O_3$ to $NH_4Cl$ reaction ratio and chlorination time on the particle characteristics of $NdCl_3$. Table 4 summarizes the particle size characteristics of the samples prepared with raw $Nd_2O_3$ and the various chlorination conditions, where three distribution points (d10, d50, and d90) and specific surface area values are presented. d90 was measured as varying from 10 to 48 μm depending on the chlorination treatment conditions. Figure 9 shows the particle size distributions of the chlorinated samples, with raw $Nd_2O_3$ represented as a dotted plot. The particle size distribution increased with an increasing reaction ratio of $NH_4Cl/Nd_2O_3$ (mol/mol) and chlorination time. The specific surface area of the particles decreased with increasing particle size.

**Table 4.** Characteristics of Nd chloride obtained from particle size distribution under various chlorination conditions.

| No. | Chlorination Time [min] | $NH_4Cl/Nd_2O_3$ [mol/mol] | $d_{90}$ [μm] | $d_{50}$ [μm] | $d_{10}$ [μm] | Specific Surface Area [m²/g] |
|---|---|---|---|---|---|---|
| $Nd_2O_3$ | - | - | 7.546 | 3.085 | 0.508 | 4.468 |
| 1 | | 9.41 | 12.865 | 5.572 | 2.056 | 2.539 |
| 2 | 240 | 11.38 | 14.076 | 5.596 | 2.073 | 2.463 |
| 3 | | 24.25 | 48.433 | 6.828 | 2.443 | 2.023 |
| 4 | | 9.41 | 10.946 | 5.376 | 2.022 | 2.613 |
| 5 | 120 | 11.38 | 11.977 | 5.473 | 2.059 | 2.552 |
| 6 | | 24.25 | 18.001 | 6.210 | 2.305 | 2.203 |

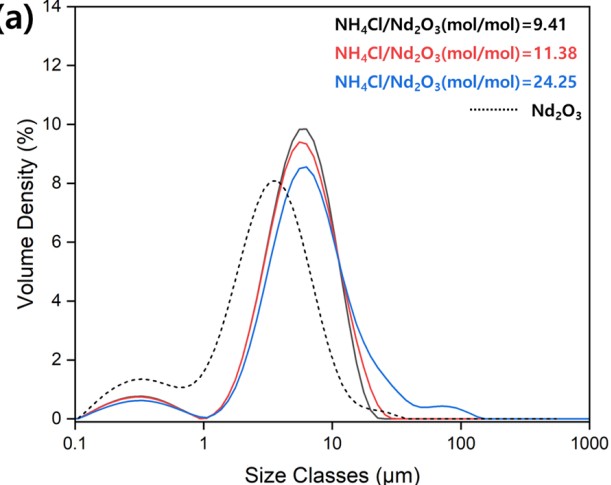
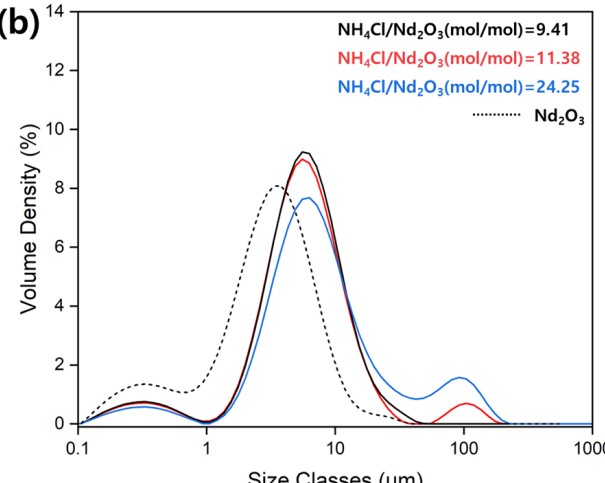

**Figure 9.** Particle size distribution of Nd chloride powder ($Nd_2O_3$ to $NH_4Cl$ reaction ratio shown in the figure): (**a**) 120 min chlorination and (**b**) 240 min chlorination.

Scanning electron microscopy (SEM) was conducted to investigate the effect of the chlorination treatment conditions on the morphology of the Nd chloride powder. The SEM images of the Nd chloride powders according to the reaction ratio of $Nd_2O_3$ and $NH_4Cl$ and the chlorination treatment time are shown in Figure 10.

The particles of the powder chlorinated for 120 min (Figure 10a–c) mostly exhibit a spherical shape with a particle size of approximately 1 μm. The fact that the ratio of $NH_4Cl$ did not affect the morphology of $NdCl_3$ powder might also mean that it did not affect the mechanism and/or rate of the reaction.

The powders chlorinated for 240 min (Figure 10d–f) had a rougher and more irregular morphology than the samples chlorinated for 120 min, and no spherical particles were observed. The agglomeration of the particles made it difficult to define the particle size, and agglomeration was observed to increase with increasing $NH_4Cl$ reaction ratio.

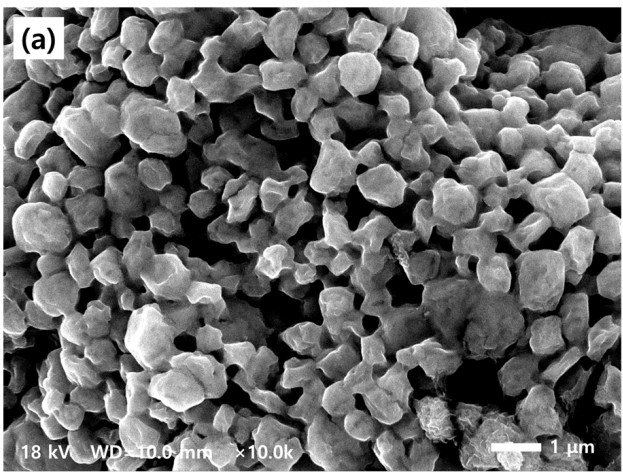
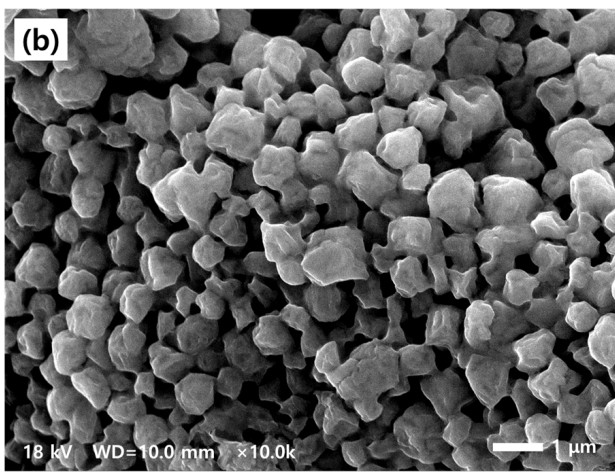

**Figure 10.** *Cont.*

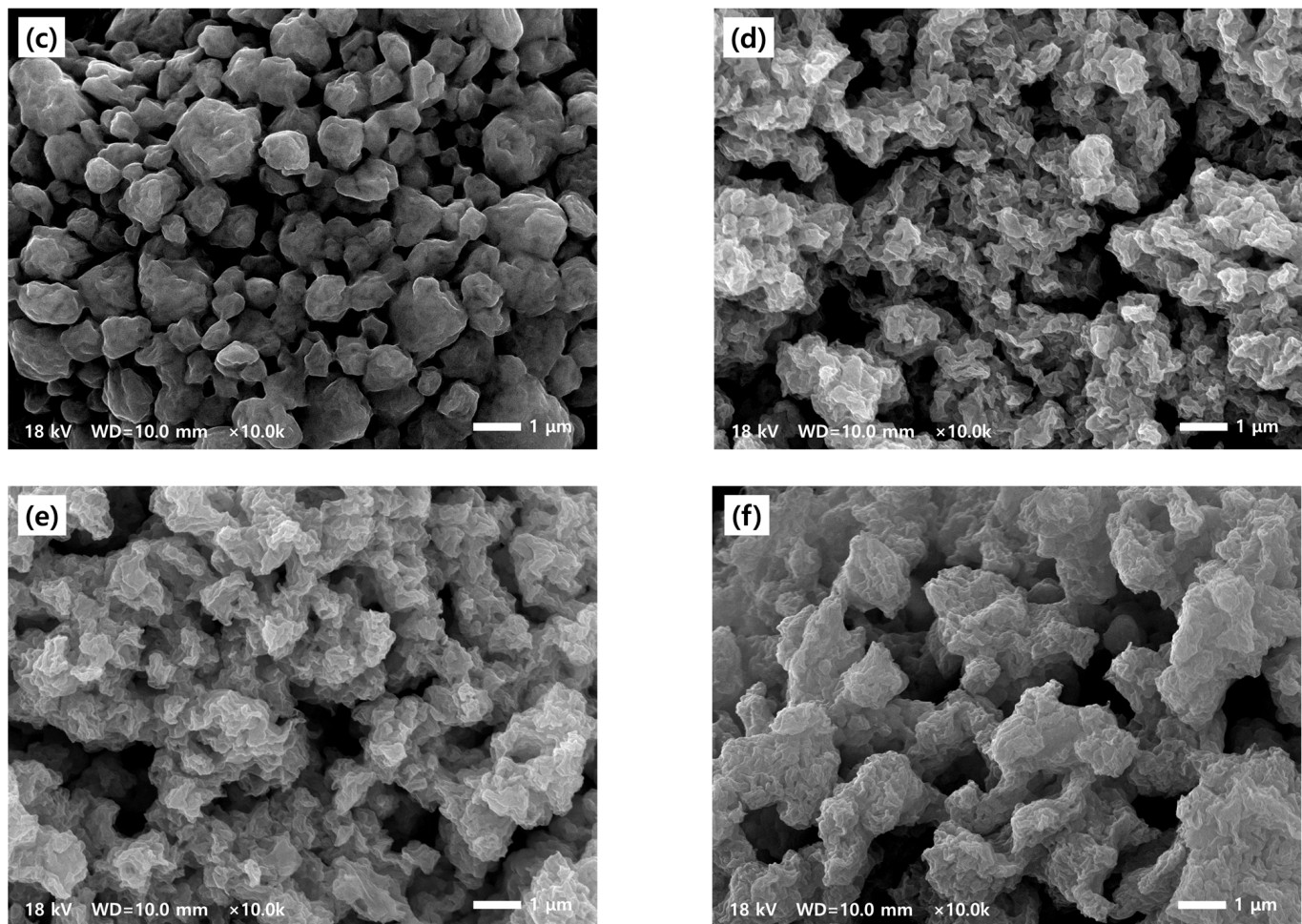

**Figure 10.** SEM images of Nd chlorides at various reaction ratios of $NH_4Cl/Nd_2O_3$ (mol/mol) and reaction times (**a**) 9.41 (mol/mol), 120 min; (**b**) 11.38 (mol/mol), 120 min; (**c**) 24.25 (mol/mol), 120 min; (**d**) 9.41 (mol/mol), 240 min; (**e**) 11.38 (mol/mol), 240 min; (**f**) 24.25 (mol/mol), 240 min.

Table 5 shows the results of the 'elemental analysis' performed to determine the concentration of impurities (N, H, and O) present in the chlorinated product; the total amount of impurities was calculated based on the mass of the product measured after the experiment. The total amount of impurities was higher in the samples chlorinated for 120 min than in those chlorinated for 240 min. The oxygen concentration and total amount of impurities in the product decreased at a higher reaction ratio of $NH_4Cl$.

**Table 5.** Elemental analysis and impurity concentration of Nd chloride powder prepared in the horizontal tube furnace.

| No. | Chlorination Time [min] | $NH_4Cl/Nd_2O_3$ [mol/mol] | Impurities [wt.%] | | | Product [g] | Impurities [g] |
|---|---|---|---|---|---|---|---|
| | | | N | H | O | | |
| 1 | | 9.41 | 0.590 | 0.626 | 1.054 | 6.9401 | 0.158 |
| 2 | 240 | 11.38 | 0.560 | 0.599 | 0.795 | 7.0211 | 0.137 |
| 3 | | 24.25 | 0.620 | 0.516 | 0.808 | 7.3434 | 0.143 |
| 4 | | 9.41 | 0.610 | 0.692 | 1.119 | 7.1062 | 0.172 |
| 5 | 120 | 11.38 | 0.740 | 0.611 | 0.998 | 7.1146 | 0.167 |
| 6 | | 24.25 | 0.640 | 0.782 | 0.814 | 7.2812 | 0.163 |

Table 6 shows that the recovery rate of $NdCl_3$ varied with the conditions, and the experimental mass of $NdCl_3$ was calculated as the difference between the masses of the product and the impurities, as listed in Table 5. No correlation between the recovery rate of $NdCl_3$ and the duration of the chlorination process was found. However, the recovery rate of $NdCl_3$ increased with higher reaction ratios of $NH_4Cl$. The theoretical production amounts of $NdCl_3$ and $NdCl_3 \cdot (H_2O)_6$ were 7.4477 g and 10.6601 g, respectively, based on 5 g of $Nd_2O_3$, and all six samples showed values close to the theoretical production amount of $NdCl_3$.

**Table 6.** Conversion rate of $NdCl_3$ prepared as powder in the horizontal tube furnace.

| No. | Chlorination | | $NH_4Cl/Nd_2O_3$ [mol/mol] | $Nd_2O_3$ [g] | $NH_4Cl$ [g] | Theoretical Mass of $NdCl_3$ [g] | Experimental Mass of $NdCl_3$ [g] | Recovery Rate [%] |
|---|---|---|---|---|---|---|---|---|
| | Temp. [°C] | Time [min] | | | | | | |
| 1 | | | 9.41 | | 7.4796 | | 6.783 | 91.07 |
| 2 | | 240 | 11.38 | | 9.0455 | | 6.884 | 92.43 |
| 3 | 400 | | 24.25 | 5 | 19.2753 | 7.4477 | 7.201 | 96.68 |
| 4 | | | 9.41 | | 7.4796 | | 6.934 | 93.10 |
| 5 | | 120 | 11.38 | | 9.0455 | | 6.947 | 93.28 |
| 6 | | | 24.25 | | 19.2753 | | 7.118 | 95.58 |

### 3.3. Horizontal Tube Furnace Apparatus: Manufacturing of Bulk $NdCl_3$

Figure 11 shows the phase composition of the bulk-shaped products manufactured by chlorination at 400 °C and subsequent heating to 760 °C, where the chlorination variables are the reaction ratio of $Nd_2O_3$ to $NH_4Cl$ (1:9.41, 1:11.38, and 1:24.25 mol/mol) and the reaction time (75 and 120 min). In all samples, $NdCl_3$ and hydrated phases ($NdCl_3 \cdot (H_2O)_6$, $NdCl_3 \cdot 5H_2O$, $NdCl_2 \cdot 4(H_2O)$, $Nd(OH)_2Cl$, and $Nd(ClO_4)_3$) were detected together, and the $NdCl_3$ phase was detected at $2\theta > 30°$ when compared to the XRD pattern of the powder form of $NdCl_3$ product. The major peaks of the samples chlorinated for 75 and 120 min with a reaction ratio of 1:24.25 were identified as the $NdCl_2 \cdot 4(H_2O)$ phase.

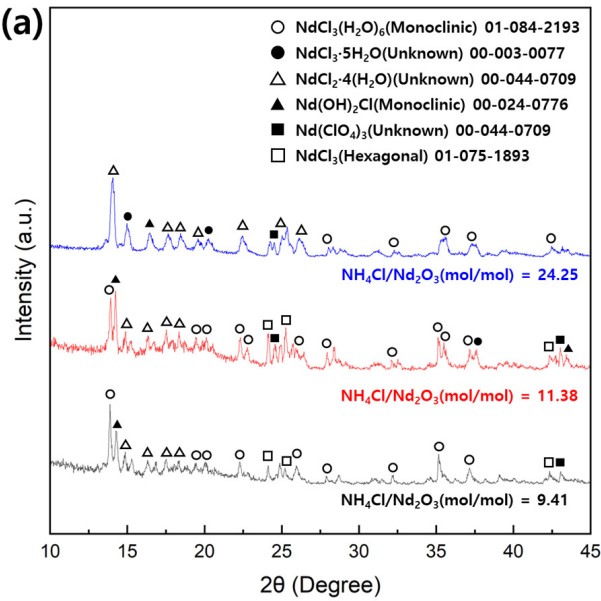 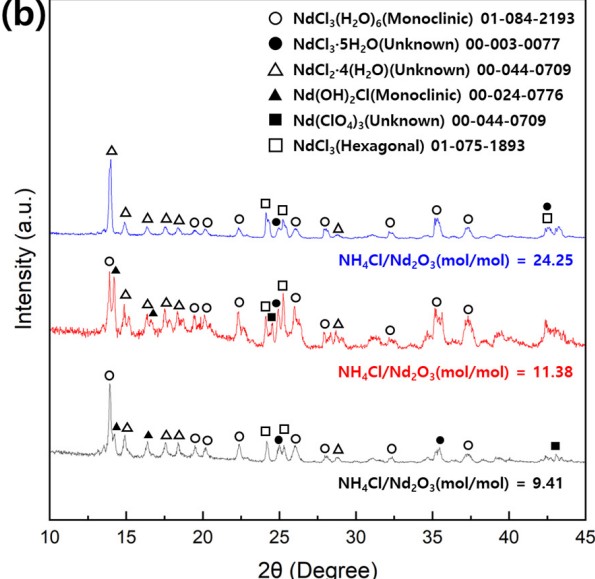

**Figure 11.** XRD pattern of Bulk Nd chloride ($Nd_2O_3$ to $NH_4Cl$ reaction ratio shown in the figure): (**a**) 75 min chlorination and (**b**) 120 min chlorination.

Table 7 shows the results of the 'elemental analysis' of the products under chlorination conditions. Results similar to those of the powder manufacturing experiments were confirmed. The total amount of impurities was higher in the sample chlorinated for 75 min than in the sample chlorinated for 120 min, and as the reaction ratio of $NH_4Cl$ increased, the oxygen concentration and total amount of impurities in the product decreased. In addition, a comparison of the products (No. 4–6) chlorinated at 400 °C for 120 min and the products (No. 7–9) melted at 760 °C after the chlorination reaction under the same conditions showed that the impurity concentration decreased overall as the product was melted.

**Table 7.** Elemental analysis and impurity concentration of bulk Nd chloride prepared in the horizontal tube furnace.

| No. | Chlorination Time [min] | $NH_4Cl/Nd_2O_3$ [mol/mol] | Impurities [wt.%] | | | Product [g] | Impurities [g] |
|---|---|---|---|---|---|---|---|
| | | | **N** | **H** | **O** | | |
| 7 | | 9.41 | 0.540 | 0.696 | 0.932 | 7.0521 | 0.153 |
| 8 | 120 | 11.38 | 0.570 | 0.505 | 0.889 | 7.0733 | 0.139 |
| 9 | | 24.25 | 0.490 | 0.481 | 0.823 | 7.2932 | 0.131 |
| 10 | | 9.41 | 0.660 | 0.479 | 1.032 | 7.0753 | 0.154 |
| 11 | 75 | 11.38 | 0.650 | 0.555 | 1.073 | 7.1461 | 0.163 |
| 12 | | 24.25 | 0.630 | 0.522 | 0.803 | 7.2032 | 0.141 |

Table 8 shows the recovery of $NdCl_3$ under different conditions, and the experimental mass of $NdCl_3$ was derived to reflect the concentration of impurities, as shown in Table 7. Similarly, no correlation between the recovery rate of $NdCl_3$ and the duration of chlorination process was found, and the recovery rate of $NdCl_3$ increased with an increase in the reaction ratio of $NH_4Cl$. In addition, the comparison of products (No. 4–6) chlorinated at 400 °C for 120 min and products (No. 7–9) melted at 760 °C after the chlorination reaction under the same conditions confirmed that melting of $NdCl_3$ product has an insignificant effect on the recovery rate of $NdCl_3$.

**Table 8.** Conversion rate of $NdCl_3$ prepared in bulk form in the horizontal tube furnace.

| No. | Chlorination | | Melting | | $NH_4Cl/Nd_2O_3$ [mol/mol] | $Nd_2O_3$ [g] | $NH_4Cl$ [g] | Theoretical Mass of $NdCl_3$ [g] | Experimental Mass of $NdCl_3$ [g] | Recovery Rate [%] |
|---|---|---|---|---|---|---|---|---|---|---|
| | Temp. [°C] | Time [min] | Temp. [°C] | Time [min] | | | | | | |
| 7 | | | | | 9.41 | | 7.4796 | | 6.899 | 92.64 |
| 8 | | 120 | | | 11.38 | | 9.0455 | | 6.934 | 93.11 |
| 9 | 400 | | 760 | 1 | 24.25 | 5 | 19.2753 | 7.4477 | 7.162 | 96.17 |
| 10 | | | | | 9.41 | | 7.4796 | | 6.922 | 92.94 |
| 11 | | 75 | | | 11.38 | | 9.0455 | | 6.983 | 93.76 |
| 12 | | | | | 24.25 | | 19.2753 | | 7.062 | 94.83 |

### 3.4. Glove Box Apparatus: Manufacturing of $NdCl_3$ Powder

Figure 12 shows the XRD pattern of the product manufactured by mixing dried $Nd_2O_3$ powder with $NH_4Cl$ at a ratio of 1:9.41 (mol/mol) and then chlorinating it at 400 °C for 120 min. The phase composition was determined to be single-phase anhydrous $NdCl_3$. A high background peak detected in the low angle region ($\theta < 20°$) occurred due to the use of an airtight holder.

SEM and PSA analyses were performed to investigate the characteristics of the $NdCl_3$ particles, and the results are presented in Figure 13.

As shown in Figure 13a, the particles exhibited irregular and agglomerated morphologies. Compared with Figure 10a (powder prepared under identical chlorination conditions in a horizontal tube furnace), the surface roughness and degree of agglomeration were higher for the particles prepared in the glove box.

Figure 13b shows the particle size distributions of the raw $Nd_2O_3$ and $NdCl_3$. The graph shows three distribution points (d10, d50, and d90) and specific surface area values for $NdCl_3$ particles, and raw $Nd_2O_3$ is represented by a dotted line. Compared with sample No. 4 in Table 4, the powder prepared in the glove box exhibited a broader particle size distribution and reduced specific surface area.

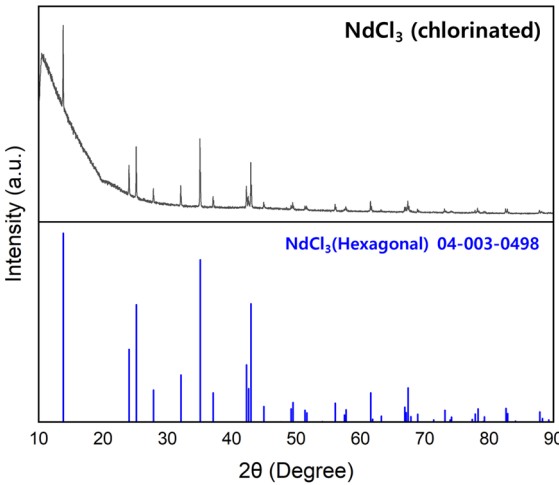

**Figure 12.** XRD pattern of Nd chloride produced in powder form in the Ar atmosphere glove box (condition: 2 h reaction at 400 °C).

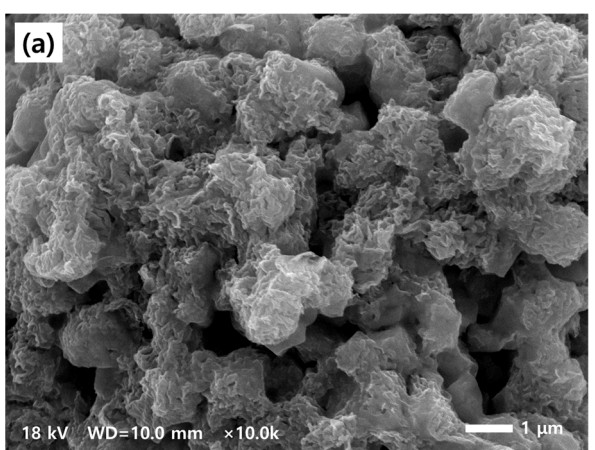
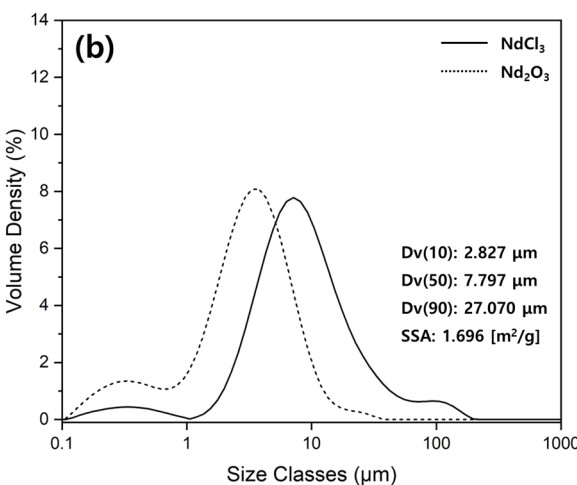

**Figure 13.** Nd chloride produced in powder form in the Ar atmosphere glove box: (**a**) scanning electron micrographs and (**b**) particle size distribution.

Table 9 shows the results of the 'elemental analysis' of the Nd chloride produced in the glove box, where it can be seen that the concentration of impurities was significantly reduced compared to the product (No. 4) produced under the same conditions in the horizontal tube furnace.

**Table 9.** Elemental analysis and impurity concentration of Nd chloride powder prepared in the glove box.

| No. | Impurities [wt.%] | | | Product [g] | Impurities [g] |
|---|---|---|---|---|---|
| | N | H | O | | |
| 13 | 0.330 | 0.349 | 0.517 | 7.4364 | 0.089 |

Table 10 lists the chlorination conditions in the glove box and the recovery rate of the produced $NdCl_3$. Considering the concentration of impurities, the recovery rate of $NdCl_3$ was 98.65%, which was the highest recovery rate compared to all products manufactured in a horizontal tube furnace.

**Table 10.** Conversion rate of $NdCl_3$ prepared as powder in the glove box.

| No. | Chlorination | | $NH_4Cl/Nd_2O_3$ [mol/mol] | $Nd_2O_3$ [g] | $NH_4Cl$ [g] | Theoretical Mass of $NdCl_3$ [g] | Experimental Mass of $NdCl_3$ [g] | Recovery Rate [%] |
| | Temp. [°C] | Time [min] | | | | | | |
|---|---|---|---|---|---|---|---|---|
| 13 | 400 | 120 | 9.41 | 5 | 7.4796 | 7.4477 | 7.347 | 98.65 |

## 4. Discussion

### 4.1. Impurity Control

The production of anhydrous $NdCl_3$ requires a reduction in the oxygen and hydrogen levels present as NdOCl and $NdCl_3 \cdot n(H_2O)$ in the product. Thermodynamic analysis suggested that calcination, chlorination temperature, $NH_4Cl$ reaction ratio, and gas atmosphere control these impurities. Calcination removes the hydrogen present as $Nd(OH)_3$ from $Nd_2O_3$, suppressing NdOCl formation during chlorination (Figure 6). Increasing the chlorination temperature enhances the driving force of the chlorination reaction but requires careful temperature control to avoid increasing NdOCl activity in the system. A higher $NH_4Cl/Nd_2O_3$ (mol/mol) reaction ratio decreased the NdOCl activity, indicating that temperature and chlorination reaction ratio variations significantly affected the $NdCl_3$ impurity levels (Figure 7).

The use of excess $NH_4Cl$ with a stoichiometric ratio higher than the reference value of 9.41 in the chlorination reaction was confirmed to reduce the oxygen concentration in the product. This finding aligns with the results of the thermodynamic analysis, which suggests that controlling the $NH_4Cl$ reaction ratio can regulate the NdOCl activity in the system (Tables 5 and 7).

The melting of the chlorinated products in a horizontal tube furnace has several implications. The reduced surface area of the products owing to melting minimizes the reaction area with ambient moisture and oxygen. Powder samples prepared from a horizontal tube furnace were confirmed to be a single-phase $NdCl_3 \cdot (H_2O)_6$, while bulk Nd chloride samples exhibited a mixture of $NdCl_3 \cdot (H_2O)_6$ and $NdCl_3$ phases (Figures 8 and 11). The formation of the $NdCl_{3(s)}$ phase can be misleadingly attributed to the decomposition of crystallization water through the melting of the product after chlorination at 400 °C. Figure 14 shows the equilibrium composition of pure $NdCl_3(H_2O)_{6(s)}$ as a function of temperature. It can be observed that $NdCl_3 \cdot (H_2O)_6$ begins to decompose at 100 °C and transforms into NdOCl above 371 °C. Considering the melting point of pure NdOCl (1100 °C) and that of the product at 760 °C, it was expected that the main phase of the samples chlorinated at 400 °C was $NdCl_3$. Additionally, the theoretical masses of $NdCl_3$ and $NdCl_3 \cdot (H_2O)_6$ that can be produced from the 5 g of $Nd_2O_3$ used are 7.4477 g and 10.66 g, respectively. The powdered and bulk samples showed close approximations of the theoretical mass of $NdCl_3$ (Tables 6 and 8). This led to the conjecture that the source of crystallization water in the bulk product was in contact with atmospheric moisture, similar to the preparation of the powder.

The gas atmosphere during chlorination was also found to be very important. The Nd chloride produced in a glove box under an Ar atmosphere was found to contain two orders of magnitude fewer impurities than the Nd chloride produced under the same chlorination conditions in a horizontal tube furnace (Tables 5 and 9). The detection of anhydrous $NdCl_3$ suggests that the regulation of oxygen and moisture is crucial during the chlorination process and also confirms that chlorinated $NdCl_3$ is very sensitive to moisture.

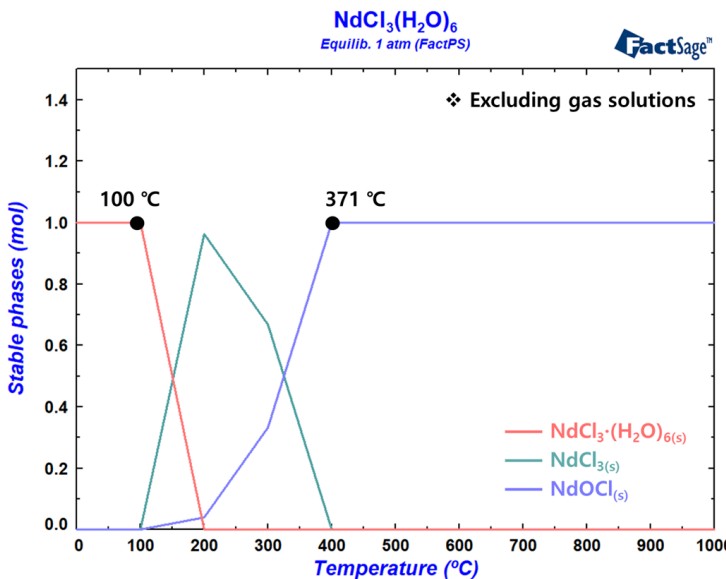

**Figure 14.** Equilibrium composition of $NdCl_3 \cdot (H_2O)_6$ for each system temperature.

### 4.2. Recovery Rate of NdCl3

An overstoichiometric ratio of $NH_4Cl$ in the chlorination process increased the recovery rate of $NdCl_3$, while the oxygen concentration and the total amount of impurities (N, H, and O) in the product decreased (Tables 5–8). This was expected because excess $NH_4Cl$ inhibited the production of NdOCl (195.69 g/mol) during the reaction while promoting the production of $NdCl_3$ (250.60 g/mol) and $H_2O$ (g). Then, it can be inferred that the residual $NH_4Cl$ was removed by sublimation after the reaction. It was considered that the lower recovery rate of $NdCl_3$ produced in the horizontal tube furnace compared to that in the glove box was caused by the continuous transport of the gaseous phase of $NdCl_3$ by the Ar gas flow from the crucible during the reaction process without reaching equilibrium with the condensed phase.

### 4.3. Particle Property

The particle characteristics of $NdCl_3$ during the chlorination process were controllable by the reaction ratio of $Nd_2O_3$ to $NH_4Cl$ and the reaction time. According to the PSA analysis, increasing the $NH_4Cl/Nd_2O_3$ (mol/mol) reaction ratio in the samples chlorinated for 120 and 240 min resulted in a wider particle size distribution and a decrease in specific surface area (Figure 9, Table 4). The SEM images show no significant difference in the particle size of the particles chlorinated for 2 h, depending on the $NH_4Cl/Nd_2O_3$ (mol/mol) reaction ratio (Figure 10a–c). The particles chlorinated for 4 h could not be defined in terms of particle size, but agglomeration of the particles was observed with increasing reaction ratios (Figure 10d–f). Therefore, it can be confirmed that the chlorination reaction time has a greater impact on particle morphology than the reaction ratio of $NH_4Cl$. The particle sizes of all the prepared Nd chlorides were larger in the PSA analysis than in the SEM images (Figures 9, 10 and 13). This indicates that the prepared Nd chloride powder has an uneven dispersion state owing to complex reasons such as water adsorption and electrostatic charges. Consequently, increasing the $NH_4Cl/Nd_2O_3$ (mol/mol) reaction ratio during chlorination can improve the thermodynamic driving force of the reaction and reduce the concentration of impurities in the product. However, further optimization of the chlorination process conditions is required to obtain $NdCl_3$ powder with precise particle size and uniform dispersion. Because the $NdCl_3$ produced in the chlorination process is used for further Nd metal production, the particle size and specific surface area of sodium chloride can act as important factors for reaction efficiency.

## 5. Conclusions

The aim of this study was to generate anhydrous $NdCl_3$ via the chlorination of $Nd_2O_3$ with $NH_4Cl$. To reduce the oxygen concentration in Nd chloride, the thermodynamic mechanisms that suppress the formation of NdOCl, an intermediate product of the chlorination process, were analyzed. Thermodynamic analysis confirmed that excess $NH_4Cl$ during the chlorination process enhanced the reaction driving force and reduced the NdOCl(s) activity in the system. As a result of chlorination, the concentration of impurities (N, H, and O) in the product decreased, and the recovery of $NdCl_3$ increased as the reaction ratio of $NH_4Cl/Nd_2O_3$ (mol/mol) increased. The $NdCl_3$ products synthesized in a horizontal tube furnace under Ar gas flow were mainly identified as hydrated phases, such as $NdCl_3 \cdot (H_2O)_6$. The formation of hydrates was attributed to the exposure to ambient air after chlorination. Based on the crystal structures and $NdCl_3$ recovery rates of the chlorinated samples, the optimal chlorination time was determined to be 120 min. An increase in the $NH_4Cl/Nd_2O_3$ (mol/mol) reaction ratio results in a broader particle size distribution and a decrease in the specific surface area. The interparticle sintering phenomenon was found to be more affected by the chlorination time than by the reaction ratio of $NH_4Cl/Nd_2O_3$ (mol/mol). Nd chloride produced in a glove box with controlled oxygen and moisture levels at the ppm level was identified as single-phase $NdCl_3$. The generated particles exhibited irregular and agglomerated morphologies. In addition, the impurity concentration was reduced by more than two times compared to the horizontal tube furnace, and the highest recovery rate (98.65%) was obtained. Therefore, the conditions for increasing the conversion rate and process efficiency of $Nd_2O_3$ to $NdCl_3$ at 400 °C are as follows: calcination of raw materials ($Nd_2O_3$, $NH_4Cl$), a reaction ratio of $Nd_2O_3/Nd_2O_3$(mol/mol) of 9.41 or higher, a reaction time within 2 h, and maintenance of Ar atmosphere. Further research on the conversion rate according to temperature and reducing agent ratio in a glove box with an Ar atmosphere is necessary.

**Author Contributions:** Conceptualization, J.-W.Y. and J.-P.W.; methodology, J.-W.Y.; software, J.-W.Y.; validation, J.-W.Y.; formal analysis, J.-W.Y.; investigation, J.-W.Y.; resources, J.-P.W.; data curation, J.-W.Y.; writing—original draft preparation, J.-W.Y.; writing—review and editing, J.-P.W.; visualization, J.-W.Y.; supervision, J.-P.W.; project administration, J.-P.W.; funding acquisition, J.-P.W. All authors have read and agreed to the published version of the manuscript.

**Funding:** This research was supported by Hyundai Motor Group, based on Contract no. 202315130001 (Development of direct reduction process technology for reducing rare metal manufacturing costs).

**Data Availability Statement:** The data are contained within the article.

**Conflicts of Interest:** The authors declare no conflicts of interest.

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
