# Peer review of "A Study on the Production of Anhydrous Neodymium Chloride through the Chlorination Reaction of Neodymium Oxide and Ammonium Chloride"

_minerals, doi:10.3390/min14050480_

Round 1

Reviewer 1 Report

Comments and Suggestions for Authors

Manuscript IDminerals-2914193

Title: A study on the production of anhydrous neodymium chloride through the chlorination reaction of neodymium oxide and ammonium chloride

Authors: Joo-Won Yu et al.

Line 31, 35, 38, 68. Authors should add references for this information.

Line 60. Avoid more than 3 references for a fact in one sentence. A maximum of 3 in a sentence is allowed for Minerals. Describe this information in detail.

Section 2.1. must be add to the section 3. Results. Authors can add some information about method for modelling (include the information both to FactSage and HSC Chemistry software) to the section 2. However, most part of this text should be relocated.

Section 2.2. Add information about raw material. Company production, grade and the particle size distribution (um) with BET (m2/g)

Authors should add the SEM images of the raw material and samples after chlorination. Authors should show shape of the different Nd phase.

How changes the particle size distribution and BET of samples compare of the different chlorination temperature, duration and Nd2O3:NH4Cl ratio? This is very important information.

Conclusions are too general; authors should indicate the underlying conclusions using paragraphs 1) 2) 3) etc. The authors should add more values to the conclusions.

The references are old, the authors should add 5-7 newer articles (from 2022-2024).

Technical errors:

Use “·” in the chemical formulas, like NdCl3·(H2O)6

Line 219, 229, 242, Figure 10 title, etc. Authors should use “h” and “min”.

Reviewer 2 Report

Comments and Suggestions for Authors

First of all, an important topic on the development of efficient and low-cost techniques for the production of carbide-based composites and coatings. However, I would suggest reconsidering the paper after inviting the authors to respond to the following comments.

1.The introduction is not enough about the background. And the introduction is confusing. It needs better orientation while writing with three steps- state of art process, research gaps and highlighting the current innovative step adopted.

2. Please give more details about the thermodynamic calculation by FactSage 8.2, for example, the activity standard state, selected solution phases and so on.

3. As Nd2O3 is one of the reactants, why not choose the Ftoxid database? Please provide the documents of all the calculations.

4. All the Figures should be drawn more beautifully.

5. References must be formatted according to journal style. Use the standard abbreviations for journal names given in the International Standard ISO 4. Some reference errors were shown in this text. Please double-check.

6. The conclusion should be concise.

7. Words in Figures 1 and 3 are too small to be seen clear. Also for most figures. Please modify them.

8. Please double-check the diffraction peak of NdCl3 in Figure 9 (a) as it looks all the same for the Nd2O3 : NH4Cl=1:11.38 and 1:9.41.

9. English expression should be modified professionally for better reading.

10. In the XRD patterns, the corresponding PDF card of the object should be marked in the picture or presented in the text.

Comments on the Quality of English Language

English expression should be modified professionally for better reading.

Reviewer 3 Report

Comments and Suggestions for Authors

The research work presents valuable insights into the extraction of rare earth metals, emphasizing efficiency and minimal environmental impact. Particularly noteworthy are the achieved outcomes concerning the precise control of intermediate products and the low levels of impurities observed in the proposed method for obtaining NdCl3.

I recommend reviewing the attached suggestions below

(1) Line 60 please check bibliographic reference 10 and 12 which do not have DOI

(2) Reference 6 rather than a study on the chlorination and fluorination of metal oxides (line 60) is an adequate reference to explain the assumptions of the thermodynamic analysis from line 80 of the article.

(3) in the paragraph that begins on line 91, it should again point out that the analysis is for equation 3.

(4) the expression of the reaction quotient, put it in a separate line from the equilibrium constant and also put the subscript of the equation to which it belongs.  (line 128)

(5) in Table 1 indicate the origin of the parameters (calculation software) (line 140)

(6) (line 142) indicate the source of the materials for the study, Nd2O3 powder, NH4Cl (reagent quality)

(7) it is desirable to give the characteristics of the equipment used, in particular XRD and TGA.

(8) line 354……….Crystallization water:line 354 -  change to : crystallization water

Round 2

Reviewer 1 Report

Comments and Suggestions for Authors

The authors have made significant revision to the article. All comments have been corrected. The article can be accepted in the present from.

Author Response

Thank you for your valuable review.

Reviewer 2 Report

Comments and Suggestions for Authors

I think the authors have taken into consideration the most comments and suggestions by the reviewers of the original manuscript and attempted to address them. However, some new questions also have existed and the quality of this paper must be improved. The related suggestion is as follows:

 1.     aand bshould be marked in Figure 13 as well as in the caption.

2.     This manuscript analyzes the chlorination mechanism of neodymium oxide for the production of anhydrous neodymium chloride. Although this study can explore the manufacturing process from an environmental perspective, this journal's " Minerals" articles should contain mining-related research findings. Therefore, the author should give an analysis of the topic of this paper and the journal's subject areas to give readers more meaningful findings.

3.     It is better to include a thermodynamic analysis of the reactivity of chlorination reactions, which should give a way forward.

4.     How the authors have decided on the temperature, time, and gas atmospheres for this experiment is not logical and clear.

5.     Please include a cost-benefit analysis. Without any financial background, this proposal for the production of anhydrous neodymium chloride is of little importance.

6.     The English of this manuscript is not sufficient for a scientific publication and must be improved professionally.

7.     Please discuss the positive and negative aspects of the state-of-the-art of production of anhydrous neodymium chloride in the introduction part. Please explain how the proposed method in this study is superior as compared to the other methods.

Comments on the Quality of English Language

The English of this manuscript  must be improved professionally.
